# Real-Time Monitoring Method for Layered Compaction Quality of Loess Subgrade Based on Hydraulic Compactor Reinforcement

**DOI:** 10.3390/s20154288

**Published:** 2020-07-31

**Authors:** Tianyu Xu, Zhijun Zhou, Ruipeng Yan, Zhipeng Zhang, Linxuan Zhu, Chaoran Chen, Fu Xu, Tong Liu

**Affiliations:** 1School of Highway, Chang’an University, Xi’an 710064, China; xutianyu@chd.edu.cn (T.X.); yanruipeng@chd.edu.cn (R.Y.); zhangzhipeng@chd.edu.cn (Z.Z.); zhulinxuan@chd.edu.cn (L.Z.); chenchaoran@chd.edu.cn (C.C.); xufu@chd.edu.cn (F.X.); 2School of Science, Xi’an University of Architecture and Technology, Xi’an 710055, China; liutong@xauat.edu.cn

**Keywords:** monitoring method, peak acceleration, layered compactness, hydraulic compactor, loess subgrade

## Abstract

Hydraulic compactor is an efficient reinforcement machine for loess subgrade. However, it is difficult to control the layered compaction quality of the subgrade. This research presents a real-time layered compactness monitoring method for hydraulic compactor reinforcement of subgrade in loess areas. The hydraulic force coefficient is first introduced, and the dynamic response model of the hydraulic rammer and soil is established. The relationship between the acceleration of the hydraulic rammer and the compactness of subgrade is then obtained based on the collision theory in elastic half space. A full-scale test using a hydraulic compactor to reinforce loess subgrade was also carried out. Results show that the hydraulic compactor obtains the effective influence depth for the reinforcement of loess subgrade. Within the effective reinforcement depth, the relationship between the peak acceleration of the rammer and the layered compactness of subgrade can be well fitted by a quadratic function model. The layered compactness of the subgrade and the working state of the hydraulic compactor can then be remotely monitored at a mobile terminal in real time. Furthermore, the monitoring technology was applied to Huangling-Yan’an Expressway in China, significantly improving the accuracy and efficiency of real-time monitoring of the layered compactness of subgrade in the loess area.

## 1. Introduction

Loess is widely distributed in the northwest of China, covering approximately 6.6% of the land in China [1,2]. The loess subgrade is predominantly comprised of filled in-layers [3]. As lack of compaction leads to uneven settlement and pavement cracking, mechanical equipment is required to reinforce the loess [4]. However, no adequate methods for monitoring layered subgrade compaction quality are available.

Determining the compactness of compacted material according to the change in acceleration of compacting equipment is a commonly utilized method. The theoretical basis is provided by investigating the dynamic response between compacting equipment and compacted material [5,6,7,8]. Previous studies have utilized a similar principle of regarding soil as an equivalent set system of elasticity and elastoplastic in which the dynamic action is simplified as a force system. The equation between two systems is then established through dynamic response, and the initial state of the object is used as the boundary condition to solve this equation [9]. The main difference is the compaction equipment utilized. Numerous scholars have determined the dynamic response between vibratory or impact roller compactors and soil [10,11,12,13,14], while others have analyzed the dynamic response between the impact rammer and soil concerning dynamic compaction of subgrade [15]. However, minimal research has explored the dynamics of the hydraulic compactor, an efficient and omni-directional reinforcement machine [16]. Although scholars obtained interesting results from the dynamic compaction impact rammer and established a dynamic model between the hydraulic-rammer and soil, the hydraulic force when the rammer comes down was not taken into account [17]. Researchers have also established a discrete element model of the hydraulic compactor reinforcing soil through numerical simulation, but this work only focuses on the rammer or the dynamic response of soil individually, rarely combining the two [18].

Experimental research into the relationship between the dynamic response of the hydraulic compactor and the compaction state of soil remains lacking. However, the experimental research methods of vibratory roller compactor-soil model and some experimental methods adopted in the development of compaction meter provide guidance for this work [19,20,21]. The aim of this investigation is to combine dynamic sensing technology with the traditional compaction measurement method. Acceleration sensors are installed on mechanical parts to collect dynamic signals, and the processed signal values are numerically linked with the monitoring indicators obtained from traditional experiments [22,23]. The relationship between roller acceleration and soil dynamic elastic modulus has been determined previously by many researchers [24,25,26,27], and a number of scholars have established the relationship between roller acceleration and compaction meter value (CMV) and compaction control value (CCV) [28,29,30,31]. Other works have also utilized the ratio of fundamental component to harmonic component in acceleration signal to reflect the compactness of soil. In other geotechnical engineering real-time monitoring fields, many scholars have used fiber optic sensors (FOS)s based on bending loss and Fiber Bragg Grating (FBG)-based FOSs to monitor structural cracks, landslides, surface subsidence, soil nails and anchors strain, pipes and pipelines strain, etc. [32,33]. The FOSs based on bending loss have relatively lower sensitivities in wide measurement ranges, but are simple to construct and low cost, without any special fibers or complex assembly. The advantage of FBG-based FOSs is that they can realize quasi-distributed measurement with high sensitivity and precision. However, the demodulation technology and cost of FBGs are relatively expensive compared to that of optical fiber [34]. In the aspect of tunnel engineering, the main monitoring directions include the stress of the adjacent tunnel lining during close excavation, and the induced settlement of the tunnel under the action of new support systems and in high-cold areas, etc. [35,36,37,38]. The above methods are accurate, and can be used for reference when studying the real-time monitoring of the compaction quality of the hydraulic compactor despite some differences. In the aspect of subgrade compaction quality monitoring, the above research illustrates that scholars have mainly established the relationship between roller acceleration and the surface compaction quality index of subgrade when a roller compactor compacts subgrade. However, the effect of the roller on the lower soil layer is seldom considered in evaluating compaction quality. As a deep compaction machine, the hydraulic compactor is beneficial to improve the overall compaction quality of subgrade if the effective reinforcement depth and the layered compactness of subgrade are guaranteed [39].

This paper establishes a dynamic model of the hydraulic-rammer and soil, determining the relationship between the acceleration of the rammer and the compactness of soil. A full-scale indoor test of hydraulic compactor was then undertaken to reinforce loess subgrade. Furthermore, the effective reinforcement depth of the hydraulic compactor is analyzed. The peak acceleration of the rammer is then used to judge the layered compactness of subgrade in the range of effective reinforcement depth.

## 2. Methodology

### 2.1. Model of Rammer Impacting Soil Mass

The process of dynamic compaction to reinforce loess subgrade is shown in Figure 1a. In the process of impact, the rammer is subjected to air resistance and friction by soil particles, resulting in some ramming energy loss and elastic-plastic deformation of the subgrade after ramming [15]. The velocities of the rammer and the center of vibration soil before collision are *v*_11_ and *v*_21_, respectively, and the velocities after collision are *v*_12_ and *v*_22_, respectively. According to the theory of collision, the whole process can be regarded as an incomplete elastic collision, which can be expressed as:(1)$$m_{1}v_{11}+m_{2}v_{21}=m_{1}v_{12}+m_{2}v_{22}.$$

The coefficient of collision recovery *e* can be expressed as:(2)$$e=(v_{22}-v_{12})/(v_{11}-v_{21}).$$

Equations (1) and (2) can then be combined to obtain:(3)$$v_{12}=v_{11}-m_{2}\left(1+e\right)\left(v_{11}-v_{21}\right)/(m_{1}+m_{2}).$$

By comparing the dynamic compaction method, the hydraulic compaction method can accelerate the fall of the rammer during the reinforcement of loess subgrade under the combined action of gravity and hydraulic acceleration system, and contains initial velocity when the rammer falls. Therefore, according to the dynamic model of dynamic consolidation method for strengthening loess subgrade, the hydraulic pressure is equivalent to the free-falling process with height of *h*. Thus, the accelerated falling process is a free-falling process at (*h* + *H*) height, as shown in Figure 1b.

To transform the process of “non free- falling process” into “equivalent free-falling”, the equivalent hydraulic force coefficient *ψ* is introduced in this research. The formula can be expressed as:(4)ψ=h/H

After the equivalent process, the whole falling process is calculated according to the free-falling process, which obeys energy conservation law. The equation can be expressed as:(5)$$m_{1}{v_{11}}^{2}/2=m_{1}gh+m_{1}gH=\left(1+\psi \right)m_{1}gH,$$
according to the initial conditions, *v*_11_ = $\sqrt{2gH\left(1+\psi \right)}$, *v*_21_ = 0, substituting *v*_11_ and *v*_21_ into Equation (3) provides:(6)$$v_{12}=\sqrt{2gH\left(1+\psi \right)}(m_{1}-em_{2})/(m_{1}+m_{2}).$$

The tamping process can be divided into two stages:(1)Deformation stage: after the rammer comes into contact with the soil, the rammer collides with the vibrating soil whose mass is *m*_2_, and the speed of the rammer instantaneously decreases from *v*_11_ to *v*_12_. Meanwhile, the static surface soil obtains significant impact acceleration and velocity. When it accelerates to the same speed as the rammer, it slows down along with the rammer until the velocity decreases to zero. In the process of rammer impact, there is a high concentration of stress, and the dynamic stress is transmitted to the soil at a fixed speed. The entire process follows the rules of conservation of momentum.(2)Rebound stage: when the soil is compressed, rebound deformation will appear, which will cause an upward rebound of the rammer and the impacted soil. With the increase of tamping times, the compactness and stiffness of the soil rises, and the rebound of the rammer is increasingly more obvious. Han Yunshan et al. [40] proposed that an effective impact stroke of the rammer includes subgrade plastic deformation and a small amplitude rebound deformation of the rammer. The subgrade deformation corresponding to the small rebound of the rammer is elastic deformation, and the deformation direction is upward, while the force of the rammer on the subgrade under the rammer is downward. This makes the rammer work negatively on the subgrade, and the degree of compaction is relatively reduced. However, the rebound process is relatively small and is negligible compared with the rammer displacement, so the rammer displacement is approximately equal to the rammer displacement of the soil on the surface of the subgrade.

### 2.2. Solution of Dynamic Balance Equation of Rammer

The compaction model consisting of half space, surface, and mass blocks described in the previous section has infinite degrees of freedom. Considering free vibration, it has a number of “vibration modes”. Gutiwiller analyzed the transient mixed boundary value by using the mathematical analysis method, solving it using the orthogonal polynomial method [41]. In this research, the instantaneous vibration is approximately described by a “basic mode” and the possibility of simulating this problem with a single degree of freedom model of equivalent set is theoretically proven. The rammer-subgrade system after impaction is simplified into a damping-spring system, and the contact between the impacted soil and the rammer with soil stiffness *k* and soil damping *c* is replaced. According to the previous study [24,30,42], soil stiffness *k* and soil damping *c* can be calculated as:(7)$$k=4Gr_{0}/(1-u),$$
(8)$$c=\pi {r_{0}}^{2}\sqrt{G\rho }/(1-\mu ),$$
where *ρ* is the density of soil, *μ* is Poisson’s ratio, and *G* is the shear modulus of soil.

The hydraulic rammer and soil move vertically. Assuming that the vertical displacement of the rammer is *x*(*t*), its speed is $\dot{x}\left(t\right)$, its acceleration is $\ddot{x}\left(t\right)$, reaction force is *F_k_*, the damping force is *F_c_*, and the resultant force of the rammer is *F*, where *F_k_* = −*kx*, *F_c_* = −*c*$\dot{x}\left(t\right)$, and *F* = −$m_{1}\ddot{x}\left(t\right)=kx\left(t\right)+c\dot{x}\left(t\right)-m_{1}g\left(1+\psi \right)$.

The equation satisfied by resultant force *F* can also be expressed as:(9)$$\ddot{x}\left(t\right)+(c/m_{1})\dot{x}\left(t\right)+(k/m_{1})x\left(t\right)=g(1+\psi ).$$

By solving differential equations, the following formula is obtained:(10)$$x\left(t\right)=e^{-\xi \omega_{n}t}\left(A_{1}\sin\omega_{d}t+A_{2}\cos\omega_{d}t\right)+m_{1}g\left(1+\psi \right)/k,$$
where *A*_1_ = (*v*_12_ − *cg*(1 + *ψ*)/2*k*)/*w_d_*, *A*_2_ = −*m*_1_*g*(1 + *ψ*)/*k*,$\omega_{n}=\sqrt{k/m_{1}}$,$\;\omega_{d}=\omega_{n}\sqrt{1-\xi^{2}}$, and *ξ = c/*2$\sqrt{km_{1}}$, $\omega_{n}$ is the undamped natural vibration frequency of the rammer,$\;\omega_{d}$ is the damped circular frequency of the rammer, and *ξ* is the damping ratio.

The second derivative of Equation (10) is thus obtained, and the acceleration time history relationship of the rammer can be expressed as:(11)$$\begin{array}{l} \ddot{x}\left(t\right)=e^{-\xi \omega_{n}t}\{\left[A_{1}\xi^{2}{\omega_{n}}^{2}-{\omega_{d}}^{2}+2A_{2}\xi \omega_{n}\omega_{d}\right]\sin\omega_{d}t\\ +\left[-2A_{1}\xi \omega_{n}\omega_{d}+A_{2}(\xi^{2}{\omega_{n}}^{2}-{\omega_{d}}^{2})\right]\cos\omega_{d}t\}=f(\omega_{n},\omega_{d},\xi ,t). \end{array}$$

Equations (7) and (8) show that *ω*_n_ and *ω*_d_ are closely related to the shear modulus *G* of soils. The shear modulus can be calculated as:(12)G=E/2(1+μ),
where *E* is the elastic modulus of soil, which varies with the number of tamping times in the course of tamping. Considering that there is no available rule for elastic modulus and tamping time of filling subgrade, Jiahuan Qian’s empirical formula is utilized in this research [43]:(13)$$E_{N}=E_{0}\times N^{0.516},$$
where *E_N_* is the dynamic modulus of elasticity after certain tamping times and *E_0_* is the initial modulus of elasticity. Thus, the acceleration relation can be expressed as:(14)$$\ddot{x}\left(t\right)=f(E_{N},\mu ,\xi ,t).$$

According to the above analysis, there is a functional relationship between the impact acceleration of the rammer and the dynamic elastic modulus *E_N_* of soil, and the dynamic elastic modulus *E_N_* of soil is an important index to measure the compactness of loess subgrade [24]. Moreover, the impact acceleration of the rammer is related to Poisson’s ratio *μ* and damping ratio *ξ* of soil. The impact of the hydraulic rammer on subgrade under the combined action of hydraulic pressure and gravity is essentially a collision between the rammer and subgrade. The impact energy of the rammer is predominantly distributed within the effective reinforcement depth of subgrade. With the increase of the energy absorbed by the soil, the displacement of the rammer in the soil continues to increase, leading to the continuous compaction between the particles of the reinforced subgrade soil, the decrease of porosity, and the gradual increase of the density and stiffness of the soil, thus the force of the reaction of the subgrade to the rammer also increases [44]. The above analysis shows that the impact acceleration moment of the rammer reflects the compact state of the subgrade, which provides theoretical support for monitoring the layered compaction quality of subgrade with the acceleration index of the rammer.

## 3. Experimental Testing

### 3.1. Testing site and Materials

To ensure the test most accurately reflects the actual situation of the project, the test material was obtained at Huangling-Yan’an Expressway, as shown in Figure 2 [23]. The full-scale model test of the hydraulic compactor reinforcing subgrade was carried out in a laboratory. The size of the model test site was 108 m long, 10 m wide, and 2.5 m deep, providing sufficient conditions to carry out full-scale tests to achieve the same effect as the actual situation.

To determine the physical and mechanical properties of the test materials, a particle sieving test, liquid-plastic limit test, large-scale direct shear test, consolidation test, compaction test, and triaxial compression test were carried out individually on the loess fillers, as shown in Figure 3. The index values of properties are provided in Table 1, in which the data shows that the liquid limit was 34% < 50%, and the plastic limit was 19.20% > 0.73 × (34% − 20%) = 10.22%. Thus, the soil classification symbol of loess was CL, according to the USCS system. The degree of compaction was measured by sand filling method and calculated using:(15)$$K=\rho_{d}/\rho_{d\max}=\rho /(1+\omega )\rho_{d\max},$$
where *ρ_d_* is dry density, *ρ* is natural density, *ρ_dmax_* is maximum dry density, and *ω* is water content.

When obtaining the shear strength parameters, three groups of samples were taken to obtain the shear stress *τ* and shear displacement Δ*s* of the soil under the normal stress conditions of 50, 100, 150, and 200 kPa, respectively. The shear strength lines (*τ-σ*) of the three groups of tests were drawn and fitted. The intercept was cohesion, and the slope was a tangent of the internal friction angle. The average cohesion was 15.47 kPa, and the internal friction was 28.66°.

The triaxial repeated compression test used to determine the elastic modulus in this study was relatively complex. As shown in Figure 3, the test instrument was a GDS triaxial test system for unsaturated soil, which was composed of a loading system, measuring system, data acquisition system, and software control and analysis system. The instrument was able to complete static triaxial testing of saturated soil and unsaturated soil under multi stress. The test process was as follows: after the soil was dried, the soil samples were screened, four groups of soil samples (39.1 mm in diameter and 80 mm in height) were prepared according to the best moisture content (12.3%), then the prepared soil samples were saturated for 10 h. The four soil samples were divided into two groups for parallel experiments. The first group of tests was to conduct triaxial compression testing until the principal stress difference *σ*_1_ − *σ*_3_ and axial strain *ε* diagram appeared to reverse bend. When the point of axial strain was greater than or equal to 15%, the test was stopped, and the failure principal stress difference *σ*_f_ was recorded. In the second group of tests, the specimens were subjected to triaxial repeated compression, preloading was conducted before the test, and the failure principal stress difference *σ*_1_-*σ*_3_ of the first group of tests was taken. As shown in Figure 4, the slope of the two ends of the last loading and unloading hysteresis loop was the elastic deformation of the soil, *E* = Δ(*σ*_1_ − *σ*_3_)/Δ*ε*. The elastic modulus values of the two samples were 25.47 and 24.65 MPa, respectively, and the average elastic modulus *E* was 25.06 MPa.

### 3.2. Test Scheme

The aim of the test is to analyze the soil settlement and the acceleration of the rammer with the number of drops, and to verify the feasibility of using the peak acceleration of the rammer to evaluate the compaction quality of loess subgrade. In order to determine the effective reinforcement depth of subgrade, it is necessary to study the variation of subgrade compactness and soil settlement with the depth of subgrade. In the range of effective reinforcement depth, it is also necessary to analyze the relationship between the peak acceleration of the rammer and layered compactness by regression analysis to provide a real-time method to monitor subgrade layer compaction quality.

Three different working conditions were designed in this experiment. As shown in Figure 5, the rammer dropping distance in working conditions 1, 2, and 3 were 2.2, 1.6, and 0.7 m, respectively. Under each dropping distance, four ramming points were set on the surface of the subgrade, corresponding to different dropping numbers. The maximum dropping numbers from left to right were 3, 6, 9, and 12 drops, respectively. The net distance between adjacent working conditions and tamping points was 1 m, and the size of the subgrade model was 6 × 8 × 1.8 m (length × width × height) [45].

Figure 6 depicts the process of the test. The water content of the loess filler was controlled in the optimum water content range of 12.3% ± 2%. The full width of the cross section and vertical horizontal layered filling compaction method were adopted for filling. The loose pavement thickness of each layer was controlled within 30 cm [46]. Six layers were filled, the initial degree of compaction was 85%, and the filling plane size was 6 × 8 m (length × width). An HHT-66 hydraulic compactor was adopted as the reinforcing equipment, the weight of rammer was 3 t, and the bottom area was 0.49 m^2^.

According to the design conditions and dropping numbers mentioned above, tamping tests were then carried out in turn. To obtain the acceleration time history curve of the rammer and read the peak acceleration of each impact, a piezoelectric accelerometer DH131E was installed on the center of the top of the rammer. At the same time, the dynamic acceleration signal was collected using a DH5922 dynamic strain gauge. As shown in Figure 5, the nails were 17.8 cm long and were mainly used as a sign of settlement, and they were laid on the top of the subgrade. To measure the layered settlement of the soil inside the subgrade, nails were laid on the top of the other five inner layers with the same layout as the surface layers, with nine nails at each position to reduce the test error. The specific operation process was as follows: before the subgrade was filled in layers, iron nails were driven into the top surface of each layer of the subgrade, then the elevation of the iron nails were measured to be H_1_. After being reinforced by the hydraulic rammer, the subgrade was excavated in layers, and the elevation of the iron nails were measured to be H_2_. The subgrade settlement was then ΔH = H_1_ − H_2_. Moreover, to identify the layered interface and locate nails when excavating in layers after reinforcement, red ropes were tied to each nail, and white powder was sprayed on each layer of soil.

## 4. Results and Discussion

### 4.1. Displacement and Peak Acceleration of Rammer

From the analysis in the methodology, it can be seen that the displacement of surface soil can be approximately regarded as the displacement of the rammer, which is the displacement response of subgrade soil to the rammer. The displacement of the rammer is closely related to the degree of compaction of soil.

Figure 7 depicts the variation curve of displacement with the number of drops of the rammer under three working conditions. Results show that the displacement of the rammer increases with the increase of dropping numbers. Under three conditions, when the number of drops reached 12, the total displacement of the rammer was 341, 298, and 196 mm, respectively, illustrating that the dropping distance of the rammer has a significant impact on the displacement of soil. In addition, due to the differences in the dropping distance and dropping numbers, the increase rates were also varied, but the overall trend was gradually decreasing. In condition 1, the increase rate of total displacement tended to be flat after seven drops. In conditions 2 and 3, the increase rate remained almost unchanged after six drops. In other words, the increase rate of total displacement in phase II was obviously lower than in phase I. The dropping distance of the rammer under conditions 2 and 3 was lower than that under condition 1, the impact energy of the rammer was smaller, and the decreasing trend of soil displacement should be slowed down after seven drops. The reason for this is attributed to the first test of condition 1 having an effect on the soil of conditions 2 and 3, which makes the soil have a certain degree of compaction before tamping.

Using the acceleration sensor and acceleration acquisition instrument, the peak acceleration of the rammer under each condition was recorded. The peak acceleration of the rammer was then determined from the average value of the rammer at each ramming point, as shown in Figure 8a.

Figure 8b shows that the peak acceleration increased with ramming. In condition 1, the increase rate of total displacement tends to be flat after seven drops, and in conditions 2 and 3, the upward trend becomes gentle after six drops. Compared with the change of rammer displacement with the number of drops, the peak acceleration displays the same trend.

The reasons for the above regularity are determined as follows: in phase I, as the soil is relatively loose, the pore of the soil is greater, the relative slip between the soil particles is easier, and the slip distance is larger, the new equilibrium can be achieved in a short time. Under the action of the rammer, the impact energy absorbed by the soil and the stiffness of the soil both increase rapidly, leading to the increase of the reaction force on the rammer and the rapid increase of the peak acceleration signal received by the rammer. In phase II, because the soil has become denser under phase I, the deformation modulus of the soil becomes large. The effect of rammer impact on improving soil compactness is not obvious, the deformation of subgrade soil tends to be slow, and the growth trend of peak acceleration signal received by the rammer also slows down. Therefore, the displacement response of the rammer can reflect the compaction state of subgrade soil, and it is reliable to use the peak acceleration of the rammer to reflect the state of soil compaction, which is consistent with the theoretical derivation in the methodology.

### 4.2. Effective Reinforcement Depth

The compactness data of subgrade measured by sand filling method were sorted, and the increment of compactness at different depths of subgrade under different dropping numbers was plotted, as shown in Figure 9.

Figure 9 shows that the initial compactness of subgrade after initial compaction by roller is not completely the same, but all are about 85%. The reason for this difference is considered to be the sudden acceleration or deceleration of the roller during driving [14]. Considering that the degree of compaction will be improved by hydraulic compactor reinforcement, the small initial compaction difference can be allowed in the actual construction. Moreover, the relative improvement of subgrade compactness compared with the initial compactness after reinforcement by hydraulic compactor will be analyzed.

At the same depth, with the increase of dropping numbers, the compactness of subgrade increases compared with initial compaction, and the increase range within 0–6 drops is obviously greater than 6–12 drops. At different depths, in condition 1, the layered compactness increases by 14.4%, 13.3%, 11.6%, 6.2%, 4.4%, and 3.3%, respectively. In condition 2, the layered compactness increases by 10.4%, 9.3%, 7.4%, 4.3%, 3.4%, and 2.8%, respectively. In condition 3, the layered compactness increases by 7.7%, 7.4%, 6.6%, 4.1%, 3.7%, and 3.5%, respectively. Therefore, with the increase of subgrade depth, the subgrade compactness decreases gradually, and the impact of dropping numbers on the degree of compaction slowly weakens. This result is due to the vertical damping in the soil, which leads to an attenuation of the transmission of the tamping energy of hydraulic rammer with depth, and the energy absorbed by the soil decreases gradually. This finding indicates that the hydraulic compactor has a certain influence depth on the reinforcement of loess subgrade.

Referring to the Handbook of Foundation Treatment published by China Construction Industry Publishing House in 2000, when the settlement of soil is 5% of the subgrade surface settlement, the corresponding depth is regarded as the effective reinforcement depth [47]. Figure 10 depicts the settlement of the soil at different depths under three working conditions. The settlement of the soil obtained from the test was a distance of 0, 30, 60, 90, 120, 150, and 180 cm away from the surface of the subgrade. The effective reinforcement depth of the hydraulic compactor under different conditions was obtained by linear interpolation method, and the results are provided in Table 2.

Table 2 shows that with the increase of dropping distance and dropping numbers, the effective reinforcement depth of the hydraulic compactor increases gradually, but the impact of dropping distance is significantly greater than the dropping numbers.

### 4.3. The Relationship between Peak Acceleration and Layered Compactness

The impact of the rammer on subgrade is due to central collision, and the impact energy essentially distributes within the effective reinforcement depth of the subgrade. Therefore, regression analysis of the peak acceleration of the rammer and the compactness of subgrade within the effective reinforcement depth was required.

Table 2 shows that the effective reinforcement depth of condition 1 is 137–149 cm, therefore, the layered compactness of 0–120 cm was taken as the research object. The effective reinforcement depths of conditions 2 and 3 were 106–118 and 94–110 cm, respectively. As such, the layered compactness of 0–90 cm was taken as the research object. The regression curves of the layered compactness of subgrade and peak acceleration of the rammer are shown in Figure 11.

Figure 11 shows that there is a clear quadratic curve relationship between the peak acceleration of the rammer and the layered compactness of subgrade under three working conditions. The fitting intervals of peak acceleration of working conditions 1, 2, and 3 are 140–205, 90–170, and 75–130 g, respectively. The degree of compaction is between 85% and 100%, and the fitting degree R^2^ is above 0.96.

Considering the site constraints, there were only four tamping points in each condition, and the number of data sets was small. As illustrated in Figure 11, at the same depth, the functional relationship between the peak acceleration of the rammer and the compactness of subgrade was minimally affected by the dropping distance of the rammer. Therefore, the layered compactness and the peak acceleration under three working conditions in Figure 11 were normalized to obtain the curve in Figure 12. The specific process is as follows. Regression analysis of the layered compactness and peak acceleration data of 0–30 cm in three working conditions (12 groups in total) was carried out, and the relationship between the peak acceleration and the layered compactness in the depth range of 0–30 cm was obtained. Similarly, the relationship between peak acceleration and layered compactness of 30–60 and 60–90 cm were also obtained. As only the effective reinforcement depth of condition 1 reached 120 cm, the relationship between peak acceleration and compactness in the depth range of 90–120 cm was based on condition 1.

Figure 12 shows that the fitting interval of compactness is (85%, 100%) and the fitting degree R^2^ is above 0.92. The regression expression of each layer can be expressed as:(16)$$K_{i}=A_{i}{a_{p}}^{2}+B_{i}a_{p}+C_{i},$$
where *K_i_* is the layered compactness, *a_p_* is the peak acceleration of hydraulic rammer, the unit is g, g = 10 m/s^2^, and *A_i_*, *B_i_*, and *C_i_* are coefficients for each layer, as shown in Table 3.

Therefore, when the hydraulic compactor is used to reinforce loess subgrade, the peak acceleration of the rammer can be used directly to calculate the compactness of each layer within the effective reinforcement depth in real time. Furthermore, the average compaction degree of subgrade can be obtained by averaging the layered compactness in the effective reinforcement depth range according to the depth weighted, making the calculation of overall compaction quality of loess subgrade more precise. The formula for calculating the average compactness can be expressed as:(17)$$\bar{K}=\overset{n}{\underset{i=1}{\sum }}K_{i}d_{i}/d_{r},$$
where *d_r_* is the effective reinforcement depth, $\bar{K}$ is the average compactness in the range of effective reinforcement depth, and *d_i_* is the depth of each layer and is taken as 30 cm in this paper.

### 4.4. Engineering Case Analysis

According to the above analysis, in the process of the hydraulic compactor reinforcing the loess subgrade, the dropping distance of the rammer is *h*, the peak acceleration of rammer is *a_p_*, and the layered compactness and average compactness can be output in real time according to the program, as shown in Figure 13. Furthermore, the layered compaction quality of the subgrade and the working state of the hydraulic compactor can also be remotely monitored at a PC or mobile terminal in real time. The overall communication architecture and server architecture are shown in Figure 14.

The experimental research results were then used in engineering practice. A hydraulic compactor was used to reinforce the subgrade in Huangling-Yan’an Expressway, as shown in Figure 15. Random selection was used to select one point to be reinforced, dropping distance was 2.2 m, there were nine drops of continuous tamping, and the peak acceleration of the rammer was obtained following each drop. Table 2 shows that the effective reinforcement depth is 1.47 m. In the range of effective reinforcement depth, the click peak acceleration was substituted into Equation (16), and the coefficient can be found in Table 3, thus the real-time layered compaction shown in Table 4 was obtained.

To verify the accuracy of the calculated layered compactness, the subgrade was excavated every 30 cm in the depth range of 0–120 cm. The compactness was measured by sand filling method after nine drops, as shown in Table 4. Therefore, in this project, the measured and calculated layered compactness after nine drops was verified, and it was found that most of the errors were within 1%, which was acceptable. Furthermore, the average compactness of subgrade could be obtained by averaging the layered compactness in the effective reinforcement depth range according to the depth weighted. Therefore, this research was applied to the real-time monitoring of layered compactness and average compactness during the construction of Huangling-Yan’an Expressway.

## 5. Conclusions

The relationship between impact acceleration of the hydraulic rammer and compactness of loess subgrade was analyzed theoretically in this paper. The filling material of the subgrade in Huangling-Yan’an expressway of Shaanxi Province was selected as the test material. According to the 1:1 similarity ratio, the model of loess subgrade was built by layers, and a full-scale test of reinforcing loess subgrade using a hydraulic compactor was carried out. Based on the peak acceleration of the hydraulic rammer, the monitoring method of layered compactness of subgrade was discussed, and the following preliminary conclusions could be drawn:(1)This paper applied the hydraulic force coefficient to the impact model of hydraulic compactor reinforcing loess subgrade based on the collision theory. The dynamic balance equation of the rammer was established, then the relationship between rammer displacement and acceleration time history was obtained. Through the combination of experiment and theory, it is proved that the rammer acceleration can reflect the compaction quality of subgrade in real time.(2)Test results illustrate that the hydraulic compactor has an effective influence depth on the reinforcement of loess subgrade. The effective reinforcement depth was mainly affected by the dropping distance of the rammer, but less affected by dropping numbers.(3)In the range of effective reinforcement depth, the relationship between the layered compactness of subgrade and the peak acceleration of the rammer can be fitted as a quadratic function. Moreover, the layered compaction quality of the subgrade and the working state of the hydraulic compactor can also be remotely monitored at the PC or mobile terminal in real time.(4)The research results provide important information for real-time and remote monitoring of the layered compaction quality of loess subgrade, which significantly improves the construction accuracy and efficiency of the hydraulic compactor reinforcing the subgrade. However, it is necessary to carry out more field tests for different typical soil conditions to obtain more test data and make the method universal. The research team is currently carrying out on-site reinforcement testing of the gravel soil subgrade area, which will provide further valuable research results.

## Figures and Tables

**Figure 1 sensors-20-04288-f001:**
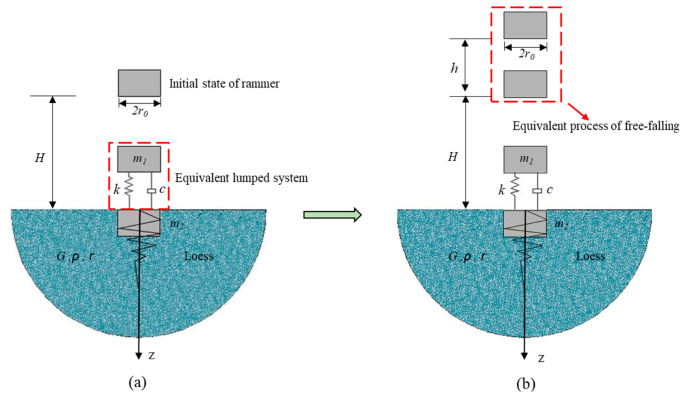
Mechanical model of rammer and soil: (**a**) dynamic compaction model; (**b**) hydraulic tamping model. For both images *m*_1_: rammer mass; *m*_2_: impacted soil mass; *r*_0_: equivalent radius of rammer; H: falling height of rammer; h: equivalent free-falling height; k: soil stiffness; c: soil damping; *G:* shear modulus of soil*; ρ*: density of soil; *r*: radius of reinforced area of subgrade.

**Figure 2 sensors-20-04288-f002:**
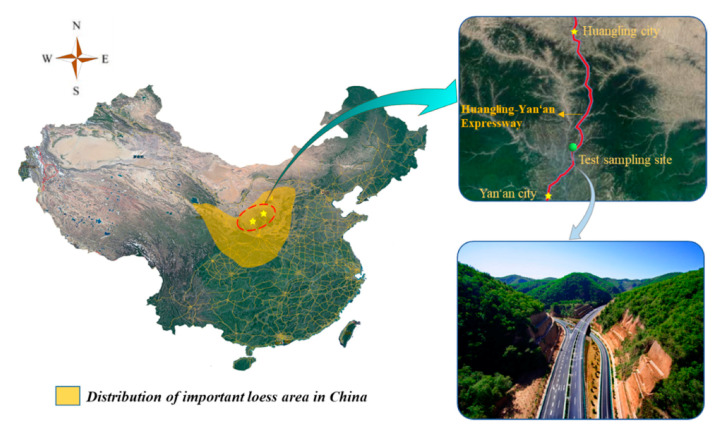
Location of research.

**Figure 3 sensors-20-04288-f003:**
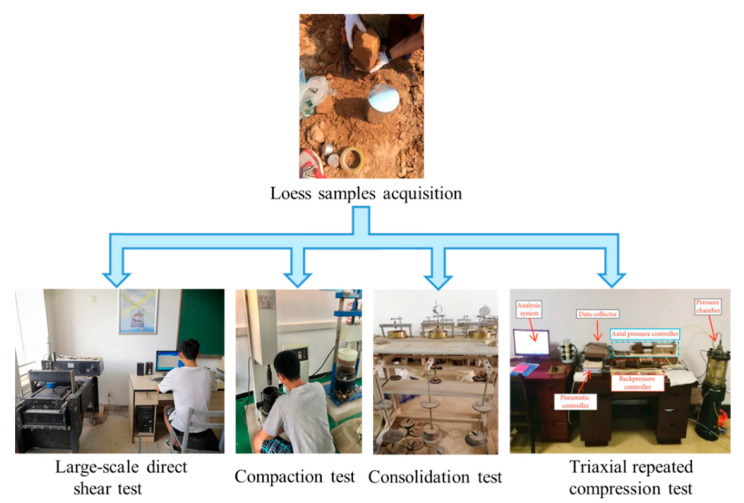
Physical properties test of loess filler.

**Figure 4 sensors-20-04288-f004:**
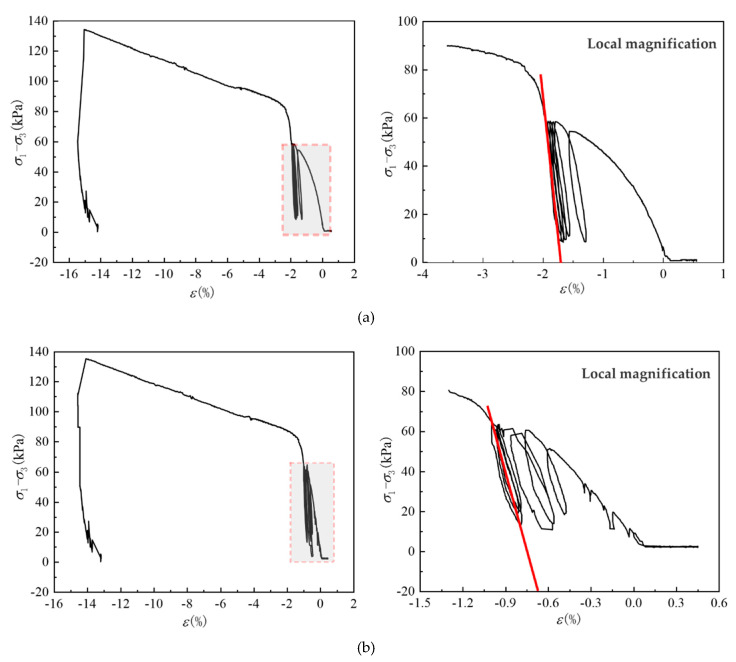
Stress strain curve of triaxial repeated compression test: (**a**) sample 1; (**b**) sample 2.

**Figure 5 sensors-20-04288-f005:**
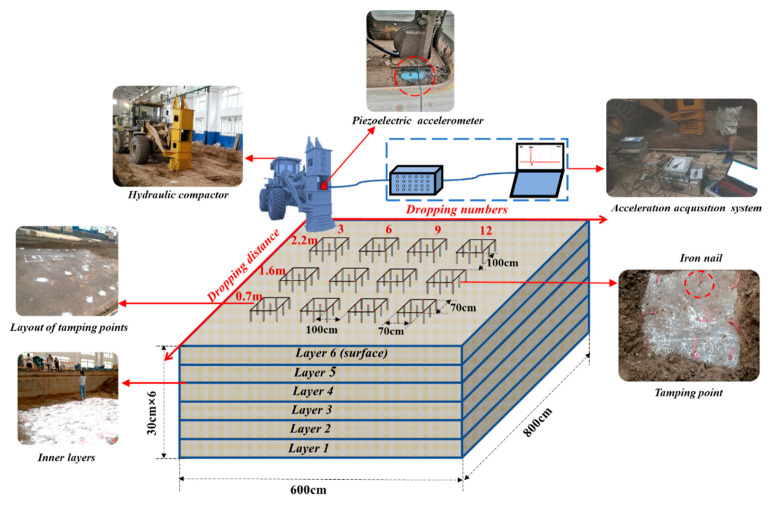
Test model and scheme design of hydraulic compactor for reinforcing loess subgrade.

**Figure 6 sensors-20-04288-f006:**
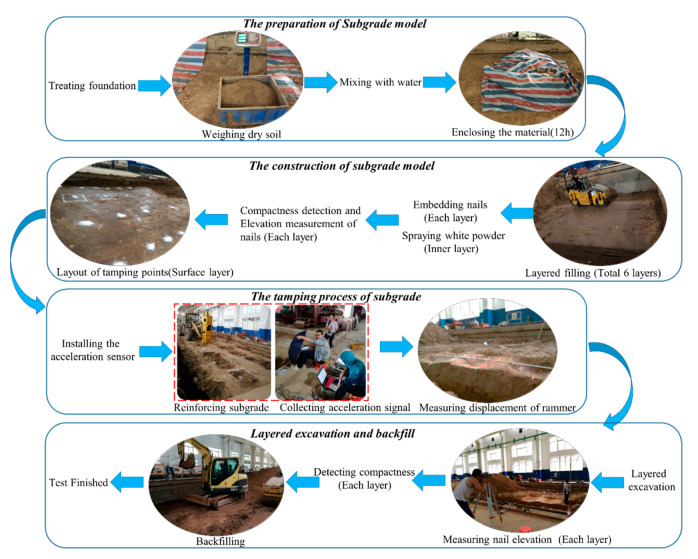
Process of hydraulic compactor testing for reinforcing loess subgrade.

**Figure 7 sensors-20-04288-f007:**
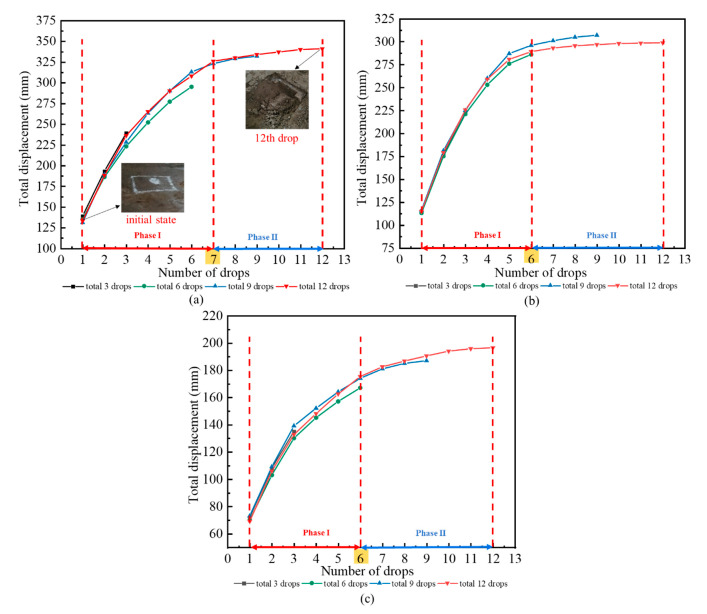
Total displacement of rammer under different conditions: (**a**) condition 1; (**b**) condition 2; (**c**) condition 3.

**Figure 8 sensors-20-04288-f008:**
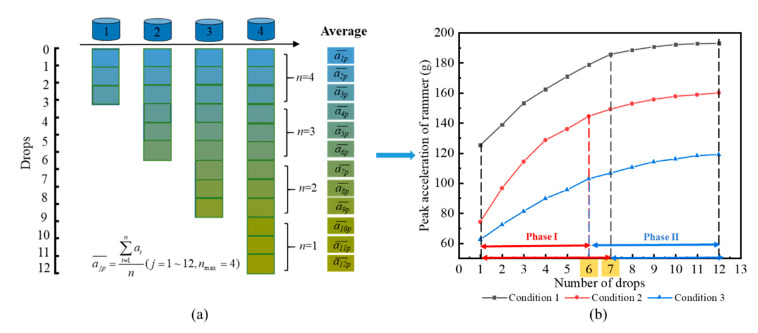
Analysis of peak acceleration of rammer: (**a**) average value of peak acceleration; (**b**) curve relationship between peak acceleration and dropping numbers of rammer.

**Figure 9 sensors-20-04288-f009:**
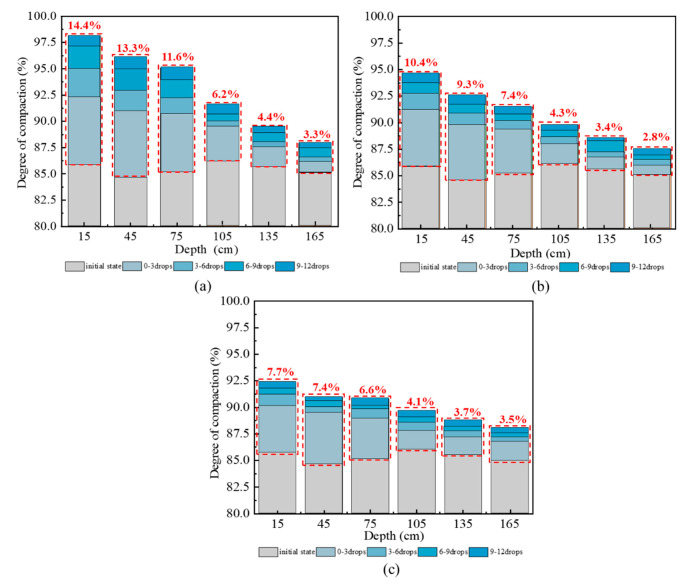
Incremental graph of layered compactness of reinforced subgrade: (**a**) condition 1; (**b**) condition 2; (**c**) condition 3.

**Figure 10 sensors-20-04288-f010:**
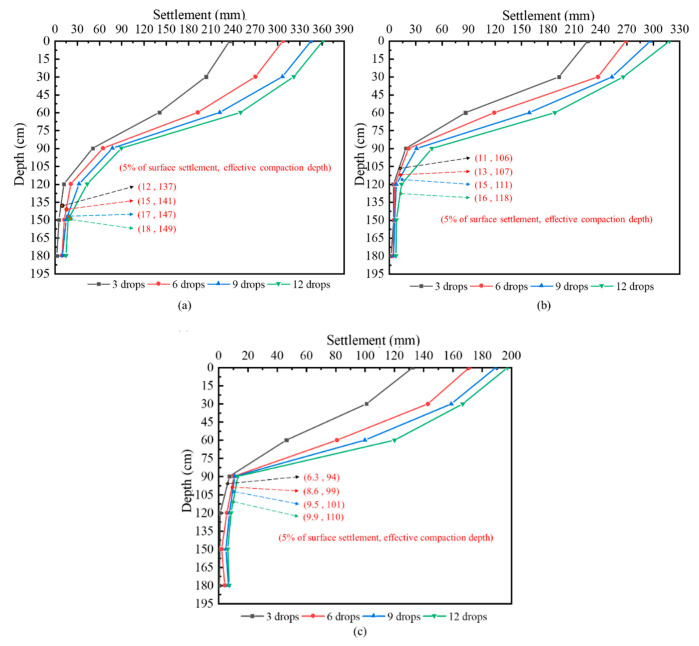
Layered settlement of soil: (**a**) condition 1; (**b**) condition 2; (**c**) condition 3.

**Figure 11 sensors-20-04288-f011:**
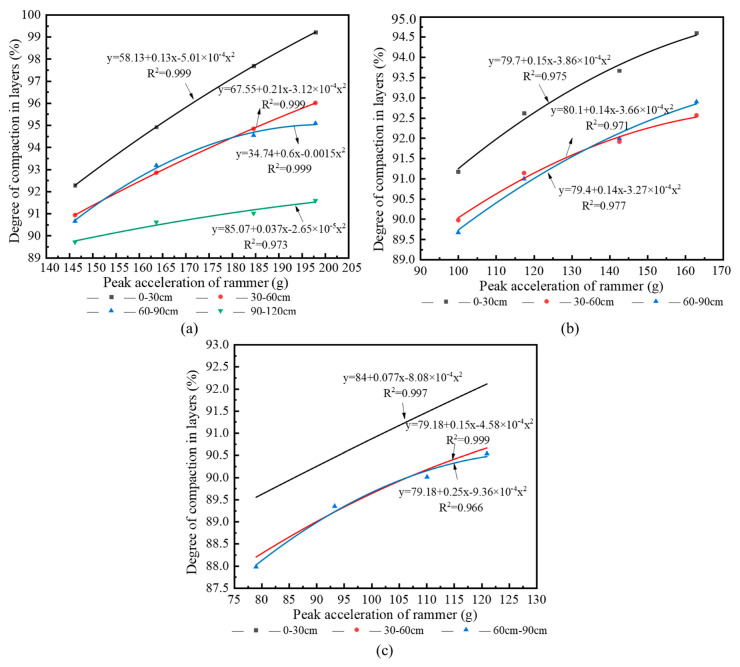
Regression curve of peak acceleration and layered compactness: (**a**) condition 1; (**b**) condition 2; (**c**) condition 3.

**Figure 12 sensors-20-04288-f012:**
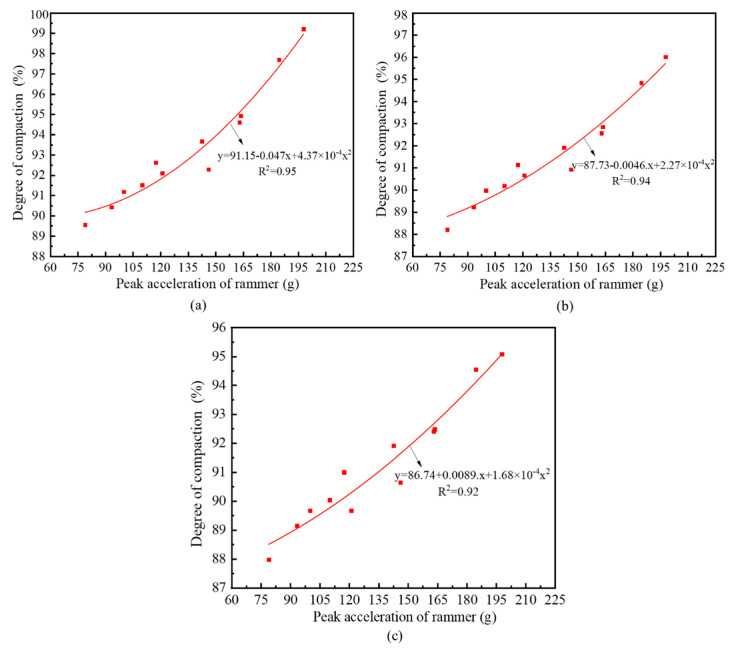
Fitting of layered compactness and peak acceleration in effective depth range: (**a**) 0–30 cm; (**b**) 30–60 cm; (**c**) 60–90 cm.

**Figure 13 sensors-20-04288-f013:**
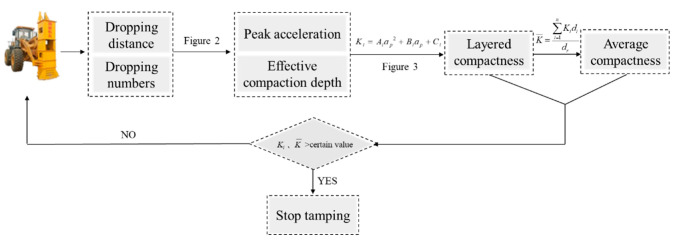
Flow chart of real-time monitoring for the compaction quality of loess subgrade.

**Figure 14 sensors-20-04288-f014:**
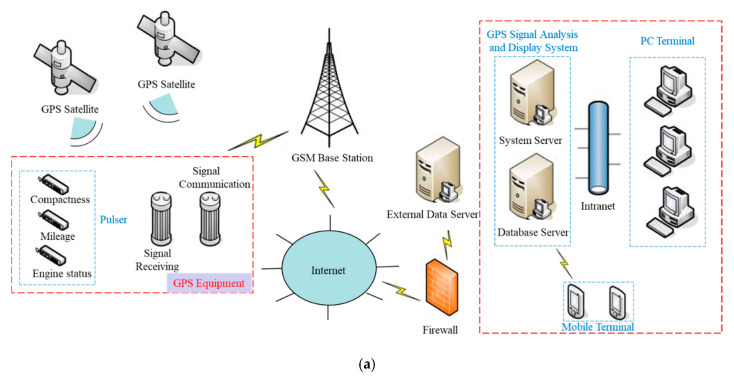
Remote real time monitoring system: (**a**) overall communication architecture; (**b**) server architecture.

**Figure 15 sensors-20-04288-f015:**
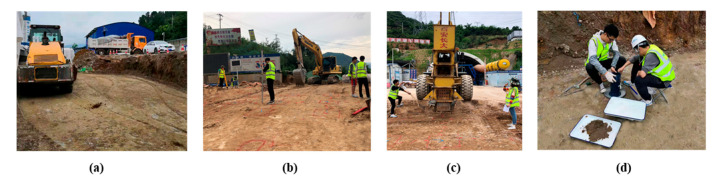
Subgrade construction of bid section-06 of Huangling-Yan’an Expressway: (**a**) layered filling; (**b**) laying up tamping point; (**c**) reinforcing subgrade; (**d**) measuring compactness.

**Table 1 sensors-20-04288-t001:** Basic parameters of model packing.

Basic Parameters	Natural Density (g/cm^3^)	Liquid Limit (%)	Plastic Limit (%)	Internal Friction Angle (°)	Cohesion (kPa)	Maximum Dry Density (g/cm^3^)	Optimum Moisture Content (%)	Modulus of Elasticity (MPa)
Value	1.62	34.00	19.20	28.66	15.47	1.90	12.30	25.06

**Table 2 sensors-20-04288-t002:** Effective reinforcement depth (cm) under different working conditions.

Dropping Numbers	DroppingDistances		
	Condition 1 (2.2 m)	Condition 2 (1.6 m)	Condition 3 (0.7 m)
3	137	106	94
6	141	107	99
9	147	111	101
12	149	118	110

**Table 3 sensors-20-04288-t003:** Regression expression coefficients.

Depth (cm)	Coefficient		
	*A_i_*	*B_i_*	*C_i_*
0–30	4.37 × 10^−4^	−0.047	91.15
30–60	2.27 × 10^−4^	−0.0046	87.73
60–90	1.68 × 10^−4^	0.0089	86.74
90–120	−2.65 × 10^−5^	0.0037	85.07

**Table 4 sensors-20-04288-t004:** Measured and calculated values of layered compactness.

Depth (cm)	Dropping Numbers	1	2	3	4	5	6	7	8	9
	Peak Acceleration (g)	96.5	122.8	142.1	158.5	168.4	172.5	178.2	182.6	186.5
0–30	**Layered compactness (%)**									
	Calculated value	90.7	91.9	93.3	94.7	95.6	96.0	96.7	97.1	97.6
	Measured value	--	--	--	--	--	--	--	--	98.2
	Error (%)									0.6
30–60	Calculated value	89.4	90.6	91.7	92.7	93.4	93.7	94.1	94.5	94.8
	Measured value	--	--	--	--	--	--	--	--	95.3
	Error (%)									0.5
60–90	Calculated value	89.2	90.4	91.4	92.4	93.0	93.3	93.7	94.0	94.2
	Measured value	--	--	--	--	--	--	--	--	93.8
	Error (%)									0.4
90–120	Calculated value	88.4	89.2	89.8	90.3	90.5	90.7	90.8	90.9	91.0
	Measured value	--	--	--	--	--	--	--	--	91.6
	Error (%)									0.7
**Average compactness (%)**		89.4	90.5	91.6	92.5	93.1	93.4	93.8	94.1	94.4

--: no data.

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
