# Peer review of "Real-Time Monitoring Method for Layered Compaction Quality of Loess Subgrade Based on Hydraulic Compactor Reinforcement"

_sensors, 2020, doi:10.3390/s20154288_

Round 1

Reviewer 1 Report

General comment: The reviewer appreciates the authors’ effort to write this paper. However, he/she suggests some corrections and rewriting.

Comment 1: Could you please explain more about why it’s called ‘Reinforcement’ in Hydraulic Compactor Reinforcement?

Comment 2: Page 7, line 227: Please elaborate more about the purpose and dimension of nail used in this study.

Comment 3: Page 2, line 82 and 85, please remove the repeated sentence: “…the whole process…can be regarded as an incomplete elastic collision.”.

Comment 4: Page 3, Figure 1, the meaning of G, ρ, r isn’t provided with.

Comment 5: Could you please provide any citation for the sentence Page 3, line 121: “The rebound makes the rammer work negatively on the subgrade, and can even loosen the compacted area”.

Comment 6: The author would like to cite recent publication of the IJGE journal to better describe damping on Page 4, line 134: Rahman, M. M., Islam, K. M., & Gassman, S. L. (2019). Correlations of permanent strain and damping coefficients with resilient modulus for coarse-grained subgrade soils. International Journal of Geotechnical Engineering, 1-10..

Comment 7: Page 5, Figure 2: why does it mean by ‘important losses’? Explain or correct.

Comment 8: How Modulus of elasticity data was obtained in Page 6, Table 1?

Comment 9: Quality of figure and axis title should be improved for Figure 5, 6, 9, 10.

Comment 10: Page 7, line 238: correct ‘Chapter 2’.

Comment 11: Page 8, Figure 6: What are the differences between Phase I and Phase II, and Condition I, II, and III?

Comment 12: Page 10, Figure 8: Is there any explanation for degree of compaction for Depth 105 cm higher than other distances?

Comment 13: Page 14, line 362: Please explain ‘certain reinforcement’.

Comment 14: Page 16, line 405: Please remove ‘creativity’.

Comment 15: Page 17, line 422, Conclusion seems more like discussion (i.e., ‘Moreover, the expression coefficients of quadratic functions at different depths were obtained, as shown in Table 3). Please rewrite the conclusion section and present the significant contribution of the paper.

Author Response

Dear Reviewer:

Thanks for the reviewer’s approval of our research. The comments from the reviewer are all valuable and very helpful for revising and improving our manuscript, as well as the important guiding significance to our research. We appreciate the input of the reviewer, and we have carefully studied the provided comments and have conducted a careful revision.We have provided a point-by-point response to your helpful comments in the attachment (PDF).

Have a nice day!

Sincerely yours,

Tianyu Xu, et al.

Reviewer 2 Report

Detailed comments: The authors proposed a real-time layered compactness monitoring method for hydraulic compactor reinforcement of subgrade in loess areas. The present topic falls within the scope of the journal and I praise the good work done by the authors. However, I cannot propose publication in its current form due not only to the poor use of English but also to some key professional issues. Following comments and suggestions would help the authors to make this paper more readable and professional.

1) Abstract: The authors declared that hydraulic compactor is an efficient reinforcement mechanism for loess subgrade. I suggest to change ‘mechanism’ to ‘manner’ or other professional terms appropriate. It obviously is not a mechanism. Further, some terms used in this section are in a lack of clear definition. For instance, what hydraulic force coefficient means? Please define them carefully in the revised manuscript. Moreover, the abstract is the summary of a manuscript. It should include the problems, methodology and main points of conclusions.

2) 1 Introduction: The authors should point out the drawbacks from previous studies in order to highlight the significance of this study. Also, the state-of-the-art researches should be compared and discussed. Below give some examples that deal with novel field monitoring techniques which can be considered in the future for geotechnical engineering projects with quality assurance demands. 1. Cheng et al., 2020. Modelling liner forces response to very close-proximity tunnelling in soft alluvial deposits. Tunnelling and Underground Space Technology, 103, 103455. 2. Zheng et al. 2020. Review of fiber optic sensors in geotechnical health monitoring. Optical Fiber Technology, 54, 102127.

3) 2 Methodology: The authors have linked EN, μ, ξ and t to the acceleration. However, it seems to me that the derived acceleration relation appear not to possess strong relevance with the following content. I am not sure why the authors want to highlight the derived acceleration relation. Further validation to confirm its relevance with the content is deemed to be essential towards making this paper more readable and logical.

4) 3 Experimental Testing: The authors conducted a series of laboratory experiments for determining the loess soil. It is noteworthy that large-scale direct shear and triaxial comrpession tests were together considered in this work to determine their shear strength parameters. Details about how the parameters can be determined should be provided in the revised manuscript. Further, the soil classification symbol, resulting from USCS system, should also be provided. Moreover, parameters shown in Table 1 include not only their physical properties but also their mechanical properties. Suggest to change the caption of this table.

5) 4 Results and Discussion: The authors declared that the first test under condition ‘1’ appears to have some influence on the soil under consitions ‘2’ and ‘3’. Can a quantitative analysis be conducted by the authors to further justify the above argument?

6) 6 Conclusions: The conclusions should be summarizing the innovative points of the new findings after research. Limitation of this study (if any) should also be stated here.

Author Response

Dear Reviewer:

Thanks for the reviewer’s approval of our research. The comments from the reviewer are all valuable and very helpful for revising and improving our manuscript, as well as the important guiding significance to our research. We appreciate the input of the reviewer, and we have carefully studied the provided comments and have conducted a careful revision. According to your helpful suggestion, our manuscript has already checked by a professional English editing service. We have provided a point-by-point response to your helpful comments in the attachment (PDF).

Have a nice day!

Sincerely yours,

Tianyu Xu, et al.

Round 2

Reviewer 2 Report

The authors seem to satisfactorily address the reviewer's comments. Suggest to publish.